# Dean-Flow Affected Lateral Focusing and Separation of Particles and Cells in Periodically Inhomogeneous Microfluidic Channels

**DOI:** 10.3390/s23020800

**Published:** 2023-01-10

**Authors:** Anita Bányai, Enikő Farkas, Hajnalka Jankovics, Inna Székács, Eszter Leelőssyné Tóth, Ferenc Vonderviszt, Róbert Horváth, Máté Varga, Péter Fürjes

**Affiliations:** 1Centre for Energy Research, Institute of Technical Physics and Materials Science, Eötvös Loránd Research Network, Konkoly Thege Miklós Str. 29-33, H-1121 Budapest, Hungary; 277 Elektronika Ltd., Fehérvári Str. 98, H-1111 Budapest, Hungary; 3Doctoral School on Materials Sciences and Technologies, Óbuda University, Bécsi Str. 96/B, H-1034 Budapest, Hungary; 4Research Institute of Biomolecular and Chemical Engineering, University of Pannonia, Egyetem Str. 10, H-8200 Veszprém, Hungary

**Keywords:** dean flow, hydrodynamic lift, microfluidics, computational fluid dynamics, lateral focusing, cell manipulation

## Abstract

The purpose of the recent work is to give a better explanation of how Dean vortices affect lateral focusing, and to understand how cell morphology can alter the focusing position compared to spherical particles. The position and extent of the focused region were investigated using polystyrene fluorescent beads with different bead diameters (Ø = 0.5, 1.1, 1.97, 2.9, 4.8, 5.4, 6.08, 10.2, 15.8, 16.5 µm) at different flow rates (0.5, 1, 2 µL/s). Size-dependent focusing generated a precise map of the equilibrium positions of the spherical beads at the end of the periodically altering channels, which gave a good benchmark for focusing multi-dimensional particles and cells. The biological samples used for experiments were rod-shaped *Escherichia coli* (*E. coli*), discoid biconcave-shaped red blood cells (RBC), round or ovoid-shaped yeast, *Saccharomyces cerevisiae*, and soft-irregular-shaped HeLa cancer-cell-line cells to understand how the shape of the cells affects the focusing position at the end of the channel.

## 1. Introduction

The very first observation of flow-generated particle positioning was made by Segere et al. in 1961, when the annual arrangement of particles was experienced in an initially uniformly diluted suspension at 0.6 times the tube’s radius, in the laminar, parabolic flow of a straight tubular channel [1].

Several channel cross-sections [2,3,4,5] were examined in the literature by different research groups over the years in order to observe the interstitial migration of particles. The main goal was to find stable lateral equilibrium positions, to reliably separate or focus different particles according to their size, to increase the distance between successive particles, and to apply these microfluidic structures for cell-level studies. By testing the flow conditions of a designed geometry, such as asymmetric curved serpentine [6,7], curved [8,9], and spiral channels [10,11,12,13], quick successes can be achieved in the field of inertial focusing with rigid particles, although in the case of biological samples, several effects complicate the process, such as the varied morphology, rigidity, and deformability of the cells; the viscosity contrast between the medium and cell interior; the shape assumed by the cell in flow; and the cell–cell interactions. Moreover, an organelle’s characteristic movement can be further influenced by the environmental effects and the characteristic properties of the chosen chip material in which the experiment takes place.

In case of the passive, channel-geometry-dependent, hydrodynamic-based separation methods, no external power is required to achieve the lateral migration of the particles. Different channel geometries are applied for enabling particle separation. In a curved, asymmetric channel geometry, the phenomenon was demonstrated by Di Carlo et al. [6] that the curvature ratio of channel geometry (δ), channel Reynolds number (*Re_c_*), particle diameter (*a*), and hydraulic diameter (*D_h_*) strictly parameterize the evoked force balance in the flow. At low Reynolds numbers, the balance of the lift forces and geometry-evoked counter-rotating Dean vortices induced Dean drag forces determining the lateral-focusing position of a given bead size in the channel cross-section.

The effect of a weak Dean flow can help lateral focusing, although in the case of the Dean drag force becoming dominant, mixing can occur instead. Accordingly, a particle size to channel cross-section ratio was defined (*a*/*D_h_* > 0.07), above which lateral focusing is possible.

The asymmetric serpentine-type microfluidic channel designs enable inertial focusing and are preferred for flow cytometry especially for high-throughput fluorescence-activated cell sorting (FACS). John Oakey et al. [14] combined an asymmetric curved channel with a high aspect ratio with a straight channel to reduce lateral focus positions to a single point. In a tortuous channel, the beads migrated laterally to one side, but in fact two equilibrium positions were formed above each other along the height of the channel. In a straight channel, on the other hand, two equilibrium positions could be isolated laterally at the same height. Competing lift forces helped to migrate the applied 10.2 µm diameter fluorescent beads to their lateral position, and hydrodynamic repulsion aided the longitudinal spacing at a flow rate of 100 µL/min. By precisely controlling the particle position and spacing at high linear particle velocities (>1 m/s), higher separation throughput and cleaner data collection can be achieved.

In medical diagnostics, efforts were made for being able to manipulate targets present in the sub-micron range, due to the size of bacterial cells being in the range of 1–3 µm. Viruses are smaller, e.g., the size of influenza viruses is around 80–100 nm [15]; the size of SARS-CoV2 ranges between 70 and 90 nm [16] (p.19). The use of passive hydrodynamic separation methods in the sub-micron (<1 µm) range is a serious challenge, since the elastic forces scale with the particle volume, and in this size range, particles are effected by the increasing phenomena of Brownian motion [17].

The recent work of Lei Wang et al. [18] has proven that this passive method is even capable to concentrate and focus particles under the size of 2 µm, such as bacteria, sub-cellular organelles, and even viruses. Fluorescent spheres were used in the size range of 2 µm–200 nm and 2 µm beads could be separated from 920 nm in asymmetric serpentine-type microfluidic channel with a channel width of 20 µm and a height of 10 µm—in the smaller curvature—at an 80 µL/min flow rate. Even 920 nm particles could be segregated from 200 nm in a tightened channel cross-section of 10 µm × 5 µm. In each case, the bigger particles became focused, while the smaller ones remained diffuse, proving the fact that the segregation of nano- and bioparticles can be possible using adequate flow parameters. The comprehensive study even detailed how sample density can degrade the degree of focusing, and how the altering target-particle rigidity affects the focusing position in continuous flow. In the case of GFP-modified cyanobacteria and spheroid fluorescent beads having the same size (2 µm), the bacteria-focusing peak was 1.05 µm closer to the centerline compared to the rigid particle, which could be explained by the shape or deformation of the biological species. The possible deterioration of biological samples due to evolving shear forces at extremely high flow rates has to be reckoned with.

Henceforth, the material of the microfluidic platform must be chosen according to the task to be performed. Polydimethylsiloxane (PDMS) is preferred by many labs for rapid prototyping because of its non-toxicity, chemical inertness, optical transparency, good adhesion for multiple substrates, cost effectiveness, and rapid and easy fabrication. Channel deformation, however, due to high-pressure application, seems to be a disadvantage. A solution may be to choose another material with greater stiffness (characterized by Young’s modulus), such as polycarbonate (2 GPa), thermoset polyester (~1.2 GPa), polyurethane methacrylate (91 MPa), or Norland Adhesive 81 (325 MPa), instead of PDMS (10:1, 2.5 MPa) to overcome the given problem areas, or at least perform the experiment on a hybrid platform: PDMS combined with glass (Pyrex Glass 63 GPa) or silicon (130 GPa) [19]. 

In this report, the measurements were executed in a PDMS-glass hybrid platform. In our previous work [20], experiments were prepared to understand how the microfluidic channel’s parametric change (height and critical width) can affect the focusing efficiency of smaller beads; what are the minimal flow rates for particle focusing of the specific parameters; and how extending the length of the curved channel would affect the focusing process. The purpose of the current report is to give a detailed explanation in the interpretation of the particle-size-dependent focusing phenomenon. On one hand, clarification needed to be obtained for why the phenomenon of focusing of smaller beads (under Ø 4.8 µm) did not occur in curved channels with square-shaped restrictions (H50 W_cr_50), and what could help to achieve the effective focusing of these Ø 4.8 and 15.8 µm spherical beads in a rectangular channel (H25 W_cr_50), interpreting the experimental results through numerical simulation. On the other hand, the aim is to examine the focusing efficiency and alterations in focusing positions when using biological particles owning different morphologies rather than the spherical polystyrene beads.

Primarily fluorescent particles with different bead diameters (Ø = 0.5, 1.1, 1.97, 2.9, 4.8, 5.4, 6.08, 10.2, 15.8 and 16.5 µm) were used in the suggested curved channel (H25_W_cr_50) at different flow rates (0.5, 1 and 2 µL/s). Due to the possible targeted cell sizes being between 0.5µm (*E. coli*) and 10–20µm (blood cells or tumor cells—CTCs), test polystyrene beads were chosen as equivalent models with bead sizes that covered this range. The flow rates were defined to ensure adequate flow conditions for inertial focusing, without deteriorating the cell membranes caused by the shear forces emerging in the case of high flow velocities. The analyzed size-dependent focusing was represented by a precise map of the equilibrium positions of the spherical beads at the end of the channel, giving a good benchmark for the behavior of the multi-dimensional particles. The lateral positions of living cells were also evaluated in relation to the lateral position of spherical particles of similar size, such as *E. coli* bacteria (rod-shaped), red blood cells (discoid-biconcave-shaped), *Saccharomyces cerevisiae* (round- or ovoid-shaped yeast), and HeLa cancer cell-line cells (soft-irregular-shaped). The focusing efficiencies were studied by fluorescent or dark-field imaging the spherical rigid particles and real biological cells having the same size range to better understand the role of particle morphology in hydrodynamic focusing phenomena.

The hydrodynamic background of particle positioning was comprehended by demonstrating the formation and migration of Dean vortex centers using finite element modelling by COMSOL Multiphysics [21].

## 2. Theoretical Background

### 2.1. Hydrodynamics of Focusing Multi-Dimensional Particles

The Reynolds number (Re) was introduced to characterize the flow in a microfluidic channel, based on the ratio between the internal friction of the medium [22,23]:(1)$$Re=\frac{u_{m}D_{H\;}\rho }{\eta }$$
(*u_m_*—max channel velocity; *D_H_*—hydraulic diameter; *η*—dynamic viscosity; *ρ*—density of the medium); and to characterize the forces acting on the particles in a closed microfluidic channel,
(2)$$Re_{p}=\frac{u_{m}a^{2}\rho }{\eta D_{H}}=\frac{a^{2}}{D_{H}^{2}}Re$$
(Re*_p_*—particle Reynolds number; *a*—diameter of the particle); in which the hydraulic diameter (*D_H_*) can be given by the Equation (3) below:(3)$$D_{H}=\frac{2wh}{w+h}$$
(*w*—channel’s width; *h*—height).

In Poiseuille flow, the “two-staged” process of inertial migration occurs due to the balance of the inertial lift ($F_{L})$, the wall-induced lift ($F_{W})$, and shear-gradient-induced lift forces ($F_{S})$ [24]:(4)$$F_{L\;}=F_{W\;}+\;F_{S\;}=\frac{(\beta^{2\;}G_{1\;}+\beta \alpha G_{2})\;\rho u_{m}^{2}\;a^{4}}{h^{2}}$$
(β—dimensionless share rate; *α*-dimensionless share gradient; $G_{1},\;G_{2}$—function of lateral position).

The equilibrium position can be further tuned by increasing the Reynolds number—particles tend to migrate closer to the walls—or by introducing a less dominant force at a low Reynolds number: rotation-induced lift force ($F_{R}$) triggered either by the shape of the particle or by geometrical considerations on the channel: (5)$$F_{R\;}\sim \;\omega \times \;u_{r\;}$$
(ω—angular velocity; *u_r_*—vectors of relative particle velocity).

In an asymmetric curved channel, a secondary rotational flow—Dean flow—can be triggered, which further reduces the lateral equilibrium positions in inertial focusing. Such a geometry was presented by di Carlo et al. [6], where the magnitude of the rotational flow velocity and the resulting drag force further tuned these equilibrium positions. The evolution of this secondary rotational flow can be characterized by a dimensionless number: the Dean number, which indicates the strength of the Dean flow, and can be calculated as [7]:(6)$$De=Re\sqrt{\frac{D_{H}}{2R}}$$
(R—the radius of the curvature).

The phenomenon evolving in the curved channel was explained in more detail in our previous publication [20]. To study the concept, rigid spherical polystyrene beads were used, which do not take on a special form of movement in continuous flow. What surrounds our curiosity is: what other effects influence the focusing positions of soft, deformable biological cells?

Jinghong Su et al. [24] presented a series of detailed three-dimensional numerical simulations studying the change in focusing equilibrium positions influenced by particle shape in straight channels having different channel cross-sections, and at altered channel Reynolds numbers (50–400). In these calculations, the non-spherical particle form factor (*L*/*D*), the aspect ratio of the particle main size to the channel height (к=L/H), the aspect ratio of the channel cross-section (*W*/*H*), and the lift coefficient ($C_{L})$—the moment of inertia tensor of the particle—were also considered when the Cartesian (overlapping) grid method was executed.
(7)$$C_{L}=\frac{F_{L}}{\frac{\rho u_{m}^{2}a_{n}^{4}}{h^{2}}}$$
($a_{n}$—normal diameter identical size between the spherical and non-spherical particle).

The equilibrium positions can be found in a given channel cross-section based on the rate of inertial migration and spatial distribution of average lift force coefficient $C_{L}$ at a steady-state condition. Therefore, equivalent bead sizes had been determined between spherical and non-spherical particles; the classification was made by: the nominal diameter ($a_{n}$), axial length diameter ($a_{a}$), and rotational diameter ($a_{r}$). These parameters optimally describe the equivalent diameter as: $a_{a}$ at Re = 50, but $a_{r}$ at Re = 200 in case of square-channel cross-sections; and $a_{a}$ at Re = 50, $a_{r}$ at Re = 100; but >$a_{r}$, when Re is higher (in rectangular channels), in that case, particle equilibrium positions are shifted closer to the centerline.

Whenever *L/D* increases, the equilibrium position of non-spherical particles is approaching the walls, although at Re = 150–200, when *L/D* = 3 and 4, this trend is experienced to be reversed.

Moreover, special movement forms can be observed, such as

Oscillation, the magnitude, and also amplitude increases with *L/D*, such as at rod-like particles;Rotation is characterized by an extreme lift force coefficient and also increases with *L*/*D*; the angular velocities (ω) are in phase; and in the case of ellipsoidal particles, ω is nearly independent of к;Tumbling motion: oblate ellipsoid in linear shear flow around the vortex axis, but with increasing Re, *L*/*D* slowly decreases.

### 2.2. Morphology and Movement of Rod-Shape Cells (E. coli) in Flow

The comprehension of bacterial motility compared to hydrodynamics-based predictions can be crucial to see the whole picture in terms of microfluidics. Motile bacteria commonly move via flagella [25] as a result of some environmental impact influenced by pH, viscosity, chemical environment, surface roughness, and so on.

Viola Tokárová et al. [26] examined five different species of bacteria (*Vibrio natriegens*, *Magnetococcus marinus*, *Pseudomonas putida*, *Vibrio fischeri*, and *Escherichia coli*), with different sizes and flagellum structures, in order to better understand their movement characteristics in chambers, linear-, angled-, and serpentine-type channels having different diameters. Cell motility was represented by density maps, spatial distribution, bacterial trajectories in 2D and 3D visualization. Based on their observations, three types of behavior were distinguished and grouped as (1) accumulators: bacteria swimming in a distance of a few tens of nanometers to the wall, interspersed with steric interactions; (2) parallel movers: hydrodynamic forces retaining bacteria in a specific region, a few micrometers from the surface; (3) escapers: the bacteria leaving the wall due to hydrodynamic interactions. The cell body aspect ratio and length of the flagellum contributes greatly to the classification.

*E. coli* has a rod-like shape, owning a high aspect ratio, and has several flagella. Based on hydrodynamic principles, and observations of species-specific movements, *E. coli* was classified as a parallel swimmer, but in a smaller proportion of cases, “wall escaper” behavior could be also identified; rarely, circular movements were also shown until they attached to the walls.

In a tortuous channel with different local channel restrictions (5, 10, 15 µm), it was observed that the hydrodynamics-driven movement was more characteristic in a wider channel, although in a narrower channel, their behavior was dominantly based on local steric interactions between the wall and the flagella. In channels characterized by a restricted region of 10 µm width, the two effects did not cooperate, parallel swimmers got easily trapped, and the ratio of the characteristic shape had a greater role. *E. coli*—which has the lowest ratio of the cell body and the flagella—could not overpass, and started to show circular movements and got easily trapped at the corners.

Based on Jinghong Su’s [24] observations in the case of *E. coli*, the cell aspect ratio *L*/*D* = 2/0.5 = 4. Based on inertial migration at *Re* = 50, the equivalent diameter compared to spherical ones could be estimated by the axial length of diameter, and the *E. coli* tended to oscillate in flow. It should be noted that, in our recent study, serpentine-type channels were used without corners, and the critical channel’s cross-sections were parameterized by 25 µm height and 50 µm width; accordingly, the hydrodynamics-driven movement of *E. coli* was dominant.

### 2.3. Morphology and Movement of Disc-Shape Cells (Red Blood Cells—RBC) in Flow

Thomas M. Geislinger et al. [27] conducted thorough research on the effects of hydrodynamic forces acting on rigid and deformable particles moving in the flow. RBCs are deformable objects, surrounded by a lipid bilayer, owning a discoid biconcave shape. In hydrodynamic flow, several motion forms of RBCs were observed: (1) tumbling motion at low shear rates, (2) tank-treading motion at higher shear rates, or (3) swinging. The difference between the viscosity of the medium and the internal cytoplasm of the cell, as well as the excess area, also influenced the characteristics of the motion. Considering the degree of shearing effect, RBCs can take on such varied forms in flow as: biconcave discoid shape, slipper shape, or parachutes.

Many questions still arise, such as how the phenomenon can be described as accurately as possible with mathematical models, and to what extent rotational lift forces contribute to the success of focusing besides shear- and wall-induced lift forces in Poiseuille flow. At a higher hematocrit level, the effective viscosity decreases with the increased shear force; moreover, the cell–cell interactions and the significant restriction of the channel cross-section (<8 µm) can, thus, drastically change the cell arrangement and dynamic shape of the cells.

The observations of di Carlo [6] showed that ordering RBCs was also possible (dilution 2 V/V%) in a straight and curved serpentine channel; however, at even at higher flow rates (*Re* = 60), cell viability was barely affected, and they all behaved as rigid particles in the flow. In a 50 µm square channel, the rotational alignment of RBCs was also observed setting the disk axis parallel to the closest wall.

Based on Jinghong Su’s [24] observations, at a cell aspect ratio *L/D* = 4:15 with a diameter of 7.5 µm and thickness of 2 µm, к in our channel should be around 7.5/25 = 0.3; its equivalent diameter compared to spherical beads can be represented by its rotational diameter, but at higher к or *Re,* the diameter can be slightly larger.

In human blood, the Reynolds number can be characterized between *Re* < 0.01, *Re*~1, and *Re* ≈ 4000 in the capillaries, arteries, and aorta, respectively [27].

Making a classification based on the shear stress, tumbling motion is characteristic at low stress and rolling motion at higher ones, but still, the stabilized cell membrane maintains its chase, even at tank-treading dynamics or off-shear-plane tumbling [28]. When shear-induced viscous stress is evolved, bouncing in circumferential directions can balance the surface stresses. In order to reduce the viscosity of the medium and cell–cell interaction, the used blood sample was also diluted in our case.

### 2.4. Morphology and Movement of Spherical Cells (Yeast and HeLa) in Flow

Eliezer Kainan et al. [29] performed inertial focusing and sorting of *Saccharomyces cereviae* in a curved channel at higher Reynolds numbers (*Re* = 215, 1.5 mL/min) to segregate yeast based on age-related cell size. The degree of budding is a good indicator of cell aging. A high-density yeast cell culture after several days showed that ~32% of the population were newborn, ~67% had 0–2 scars, ~33% had 3 ≤ scars, and less than 5% had 11 ≤ scars. Adult yeasts were classified with the size of 5.4–11 µm (3–10 scars) and the younger ones with 2.3–4.7 µm (0–2 scars). The linear growth rate was about 0.8 ± 0.1 µm/scar, the produced chitin-rich bud scars were fluorescently labelled, and the histogram of forward scatter (FCS) of yeast populations was measured from the collected yeast populations at the end of curved channel. The collection was completed with a high throughput (10^7^ yeast/min), and without any cell damage despite the quite high shear stress (691 Pa). The smaller, younger cells left the centered ports enriched with ~12%, compared to the presence of young yeasts in the original mixture, having a scar number of 1.3 ± 1.6; and the larger, adult cells left the channel at a concave port with an enrichment of ~46%, having an average budding number of 3.4 ± 1.7. PDMS-glass hybrid platforms were used for the performed test.

During our measurements, the same strains were used, and the diameter of the used yeast approximately fell between 5 and 10 µm, but the age distribution of the cells was ignored. Only their hydrodynamic behavior and their focusing position were taken into account.

## 3. Materials and Methods

### 3.1. Materials

The fluorescent polystyrene beads were obtained from various companies: Spherotech GmbH (Ø = 0.5, 1.97, 5.4, 6.08, 10.2, 15.8, 16.5 µm, respectively), Thermo Fisher Scientific (Fluoro Max Ø = 1.1 µm), COMPEL (Ø = 2.9 µm). The origin of the biological samples was as follows: GFP-expressing *E. coli* were taken from the University of Pannonia, Research Institute of Biomolecular and Chemical Engineering [30]; the HeLa cells were received from the Centre for Energy Research (EK) Nanobiosensorics Department (Horváth et al.); the RBC sample from 77 Elektronika Ltd.; and the Saccharomyces cerevisiae yeast cells, as well as the target sample medium (phosphate-buffered saline (PBS)) were purchased from Merck Life Science Ltd.

### 3.2. Fabrication of the Microchannels

Microfluidic test structures were fabricated via the soft lithography technique using polydimethylsiloxane (PDMS). SU-8 epoxy-based negative photoresist (2025/2050/2100) was applied as a molding replica for the microchannels. Silicon wafers were spin-coated and pre-baked at 65 °C and 95 °C by the Brewer Science Cee 200CBX spin–bake system. Coated wafers were patterned by UV photolithographic exposure in a Süss MicroTech MA6 mask aligner. After exposure, the pattern was developed in a Süss MicroTech spray developer and baked at 95 °C on a hotplate. The thickness of the molding replica and thus the height of the microchannel were determined by the applied SU-8 types and was set to the following levels: 25 µm and 50 µm. PDMS pre-polymer with 1:10 elastomer/curing agent ratio was poured onto the replica and cured for 90 min at 65 °C. The resulting PDMS microfluidic chip was treated by oxygen plasma using 50 W plasma power, 100 kPa chamber pressure, and 1400–1900 sscm oxygen flow in a Diener Pico plasma etcher and bonded to a microscope glass slide by low-temperature bonding.

### 3.3. Microfluidic Design Aspects

The experimental measurements were executed in a ~35 mm long periodic sequence of asymmetrically curved serpentine channels consisting of 23 curvatures with a narrower and a wider curve in the structure. The channel geometry parameters were varying as described in Figure 1.

### 3.4. Finite Element Modelling of the Structures and Biological Sample Particles

The flow behavior in the microfluidic channels was also studied through computational fluid dynamics (CFD) simulation using COMSOL Multiphysics (version 5.3a). Our aim was to analyze and predict particle movement in these structures. The Navier–Stokes equation was evaluated by the finite element modeling (FEM) solver considering 3-dimensional stationary laminar flow due to the low-Reynolds-number regime, with the following governing equations:(8)$$\rho \left(\frac{\partial u}{\partial t}+u\cdot \nabla u\right)=-\nabla p+\nabla \cdot \left(µ\left(\nabla u+{\left(\nabla u\right)}^{T}\right)-\frac{2}{3}µ\left(\nabla \cdot u\right)I\right)+F$$
(9)$$\frac{\partial p}{\partial t}+\nabla \cdot \left(\rho u\right)=0$$
where **u** is the velocity field, *p* is the fluid pressure, *ρ* is the fluid density, and *μ* is the fluid dynamic viscosity.

At the inlet, a laminar inflow boundary was applied with varying flow rates between 0.5 and 6 µL/s. Zero pressure with suppressed backflow was set as the outlet boundary condition and a ‘no slip’ condition was set for the channel walls. The material parameters were set to be similar to the mechanical properties of room-temperature water (density: 1000 kg/m^3^; kinematic viscosity: 10^−6^ m^2^/s). The maximal cell Reynolds number was calculated to be between 0.09 and 3.26, the volume average cell Reynolds number was estimated in the 0.01–0.79 range in all cases. A mesh-dependent convergency study [31] was conducted on the part of the whole microfluidic structure. A surface was selected at the cross-width of the smaller curve and the surface average vorticity magnitude (1/s) was calculated and considered as a probe point, with the results summarized in Table 1.

The result converged after mesh #5; mesh #7 was selected for use with the following mesh statistics and parameters: number of mesh elements: 2,3018,289; minimum element quality: 0.1417; average element quality: 0.6651; element volume ratio: 0.002995; and mesh volume: 1.724 × 10^7^ µm^3^.

Time-dependent particle trajectories were calculated by the particle-tracing module in the pre-solved stationary velocity field. A total of 1000 spherical particles were released with uniform distribution from the inlet surface with the projected plane grid initial position setting. The stick boundary condition was set for the channel walls. Exploiting structure periodicity, 3D geometry was built for the representative fraction (2 waves) of the microfluidic channel, and a periodic boundary condition (continuity) was applied at the outlet of the section mapping the particles back to the inlet with their last position to gain data for the full channel length. Particle properties were set to be in correspondence with fluorescently labeled polystyrene beads applied in the experimental validation (density: 1055 kg/m^3^, diameters: 0.5 µm, 1.1 µm, 1.97 µm, 2.9 µm, 4.8 µm, 5.4 µm, 6.08 µm, 10.2 µm, 15.8 µm, and 16.5 µm). The global error estimate of the model was 6.63 × 10^−10^.

### 3.5. Measurement Setup and Processing Experimental Data

The microfluidic system was driven by a syringe pump. Experimental data images were recorded by a Zeiss AxioVert A1 inverted microscope in fluorescent or dark-field mode and further processed. The relative lateral distribution of particles was calculated by considering the recorded fluorescent- or scattered-light-intensity map at a given cross-sectional plane. Distributions were then transformed and uniformized to become comparable with modeling data.

### 3.6. Data Processing of Numerical Model Results

The results of the numerical model were obtained by exporting particle coordinates and diameters for all timesteps. Full channel length particle trajectories were restored from the periodic data. A Poincaré map was calculated in Matlab from the dataset at the given y-z cross-sectional plane. The evolution of the lateral particle distribution in the microfluidic channel is presented by the movie (titled A_Banyai_LatFocus_FEM-Poincare.mp4) in the Appendix A. Particle midpoint coordinates were extended with 1440 points according to their radii, and particle distributions were calculated based on the histogram of these extended data points across the channel. Calculated distributions were normalized to ensure comparability with the optical measurements.

The individual fluidics were identified on the basis of the channel height and the critical cross-section as indicated in Figure 1. The velocity field simulations were performed in the smaller bend, while the lateral bead positions were measured in the larger bend and simulated in the 23^rd^ bend at the end of the channel.

## 4. Results and Discussions

### 4.1. Models of Dean-Flow-Affected Lateral Focusing in Rectangular Channels

The formation of Dean vortices and the movement of their center position were simulated in COMSOL Multiphysics as presented in Figure 2. The height of the obtained curved channels was considered as 50 µm; their critical width altered between 150, 100, and 50 µm; and the flow behaviors were compared to the cases characterized with the channel parameterized by the height of 25 µm and critical width of 50 µm (H25_W_cr_50), in which the best lateral separation of 4.8 and 15.8 µm beads was achieved at 0.5 µL/s flow rate [1]. Although in the case of the smaller channel cross-section (H25_W_cr_50), 2 µL/s was the maximal experimental flow rate (to prevent leakage and channel deterioration), the simulations were implemented in even higher flow rates of 0.5, 1, 2, 4, 8 µL/s, respectively.

Velocity-field-based streamlines were visualized in the y-z plane of the channel cross-section at the smaller curvature (W_cr_ + 50 µm) (see Figure 1). Based on the computational results, the lateral movement of the centers of the evolved vortices was the smallest in the case of the wider critical cross-section (150 µm), and the highest in the case of intensive channel height restriction.

In the case of H50_W_cr_150 (aspect ratio at the smallest channel cross-section—AR 1:3), between 0.5 and 8 µL/s flow rates, the vortex centers only moved ~40 µm towards the centerline of the channel, but the lateral position remained in the lower third of the small curvature.

At the same height, but smaller width, H50_W_cr_100 (AR 1:2), the vortex centers travelled ~50 µm towards the opposite side passing through the centerline of the channel. A similar observation was experienced in the case of H50_W_cr_50 (AR 1:1), where the vortex center travelled only ~35 µm. 

In the case of the structure H25_W_cr_50 (AR 1:2), compared to H50_W_cr_100, the lower and elongated vortex center traveled an even greater distance of ~60 µm, and significantly approached the opposite side wall. In the case of this channel geometry, not only did the lateral focusing of smaller beads perform better, the size-dependent separation of the particles was more effective.

### 4.2. Experimental Validation of the Lateral-Focusing Model

The experimental and computational results were compared by studying the lateral-focusing behavior of 4.8 and 15.8 µm beads in the channels H50_W_cr_50 and H25_W_cr_50 at 0.5, 1 and 2 µL/s flow rates in Figure 3.

The colored velocity maps in Figure 3 demonstrate the magnitude of the lateral components of the in-plane flow velocity distribution (y component of the velocity field): the red color denotes lateral secondary flow orienting to the right (positive y directions) while the blue color denotes the lateral movement to the left (negative y directions). While the formation of Dean vortices was clearly demonstrated in both microchannels (visualized alongside the cross-sectional velocity field denoted by arrows), in the higher channel (H50_W_cr_50—Figure 3A), it evolved more prominent and more symmetrical vortices compared to the lower one (H25_W_cr_50—Figure 3B).

In the case of the structure denoted by H50_W_cr_50, the larger beads (Ø 15.8 µm) were already laterally focused at 0.5 µL/s flow rate; however, the smaller ones (Ø 4.8 µm) could not be at even higher flow rates. In our previous experimental work, the flow rate was altered between 0.5 and 6 µL/s in this specific structure. In this flow rate range, the trajectory band of the 4.8 µm size beads narrowed until 3 µL/s, but above that, it started to broaden again. This phenomenon is in good accordance with the simulated results (see the lateral distributions in Figure 3A) and could be explained by the mixing effect of the evolving Dean vortices.

In the case of the H25_W_cr_50 structure, the numerical simulation predicted the effective focusing and size-dependent segmentation of the beads, as the experimental results also proved.

### 4.3. Focusing of GFP-Escherichia coli

Genetically engineered bacteria were used for the focusing experiments, which express green fluorescent protein and emit green light at a wavelength of 509 nm when excited at a 395 nm wavelength. The process is nontoxic, and since the fluorochrome is encoded by the amino acid sequence, there is a rather resistant type of marking, although a loss of fluorescence has been observed at temperatures above 60 °C or during strong acidification [32]. A recent study did show that, although the presence of GFP-encoding plasmids does not involve the affection of morphological, biochemical, or survival characteristics of the bacteria, fluorescence loss can occur over a long period due to plasmid loss in spontaneous segregation, a deletion in the replication and recombination process, or due to being affected by the nutrient supply. The plasmid stability may differ in strain-to-strain comparison. In some cases of *E. coli* O157:H7 strains, the total GFP plasmid loss was measurable after one day (EC1259 and EC1767), within 19 days (EC 811 and EC1628), or maintained plasmids during the time period of measurements (EC22 and EC 1513) (ref. [33], p. 7).

In the short term, the plasmid loss is not significant; accordingly, in our case, this phenomenon did not have to be considered, and it could be stated that the GFP modification did not affect the morphology of the bacteria significantly.

The *E. coli* bacterium is rod-shaped, the smallest dimension is 0.5 µm, and its length is usually around 2 µm [34]. Based on the literature review, in our chosen channel cross-section and applied high flow rates, the presence of steric interactions between the cell surface and the channel wall can be neglected; only the hydrodynamic effects need to be considered. It should be noted that compared to wild-type, the GFP-modified *E. coli* bacteria applied in our experiments are unable to grow a flagellum or swim.

The concentration degree of the bacterial flow was compared to the behavior of rigid beads having sizes of 0.5, 1.1 and 1.97 µm which were used as equivalent models (see Table 2). In the case of smaller bead sizes, the focusing criteria—a/D_h_ > 0.07—defined by Dino Di Carlo [6] could not be reached. These a/D_h_ ratios for the model beads were calculated as: 0.015 (for Ø 0.5 µm); 0.033 (for Ø 1.1 µm); and 0.059 (for Ø 1.97 µm) (as summarized in Table 3). According to the theoretical model, focusing did not occur in experimental tests within the chosen channel parameters (H25_W_cr_50). In contrast, the efficiency of lateral concentration increased with larger bead sizes—the lateral channel filling relative to the width of the large curvature decreased gradually from 100% to 86.8% to 67% for the 0.5, 1.1 and 1.97 µm bead sizes at a 1 µL/s flow rate, as demonstrated in Figure 4. Based on the presented theoretical model, lateral focusing to a single point would only be possible using spherical beads having a diameter above 2.4 µm. Among the applied model beads, the Ø 2.9 µm bead (a/D_H_ = 0.087) was left in a focused state along the 3 cm long microfluidic channel.

In accordance with the above-mentioned theoretical considerations, *E. coli* also was only concentrated in the periodically curved channel, as their lateral distribution estimated by the fluorescent intensity profile was recorded in the large bend. Based on the degree of concentration (88.6%), their behavior was similar to the profile of the Ø 1.1 µm beads in Figure 5, which suggests an oscillating movement of the rod-shaped particles in continuous flow.

### 4.4. Focusing of Red Blood Cells

The shape of a red blood cell is bicoid biconcave, the average cell size is between 6 and 8 µm [36], and its thickness can be 2.5 µm [35]. One droplet of blood was diluted 100× times and the diameter of the model fluorescent rigid beads was 2.9, 4.8, 5.4 and 6.08 µm, as presented in Table 2. As the bead size increased, the lateral position of the individual species shifted from the bigger radius bends towards the centerline of the channel. Based on its lateral focusing positions represented in Figure 5, the bead with a diameter of 6.08 µm describes mostly the equilibrium position of the RBC, which can explain its equilibrium position by the presence of rotational movement based on its larger volume.

### 4.5. Focusing of Yeast Cells

Depending on the morphology of yeast, its size falls between 5 and 10 µm in diameter [37]; in our experiments, the above-described budding effect was neglected. The target cells were dispersed in the PBS solution at a concentration of 0.5 mg/mL *m*/*v*, and diluted at 10 times the volume. The cells were examined in a Bürker chamber; the average cell size was determined to be around 9.75 µm and the yeasts had an ovoid- or round-shape morphology. Model beads with 5.4, 6.08 and 10.2 µm diameters were used as equivalent model particles (see Table 3). The above-mentioned tendency was observed during the focusing of the beads; the rigid particles with 10.2 µm diameter were focused in the lower half of the large bend, significantly further from the centerline of the channel, than the 6.08 μm beads. The larger particle hit the lateral position of the yeast in the flow with greater accuracy as demonstrated in Figure 5.

### 4.6. Focusing of HeLa Cells

Based on the literature review, the sizes of the individual HeLa cells were determined in the range of 16–29 µm [38]. In terms of cell size, the measurement can be considered risky in the tortuous channel with an internal height of 25 µm. In the case of rigid beads set as an equivalent model, the larger bead diameter used was 15.8 µm. The obtained HeLa suspension was diluted in PBS solution and the actual cell size was measured in a Bürker chamber. The average individual cell size was determined to be around 24 µm; however, among the various shapes (spherical, ovoid, and amorphous), there were aggregates whose larger dimension even reached 61 µm, so, in some cases, the fluidic chips were also blocked. During the focusing of the fluorescent model beads, a kind of tendency could be observed: the smaller beads were focused laterally in the lower third of the larger bend and shifted towards the centerline of the channel as their size increased. Based on the fluorescent intensity profile, the beads with a diameter of 15.8 µm were located above the centerline, slightly overlapping the focusing band of the HeLa cells (see Figure 5). The focusing position of the larger 16.5 µm diameter beads was different; these particles focused laterally in the lower half of the channel.

### 4.7. Morphology-Dependent Lateral Focusing in Low-Aspect-Ratio Microchannels

Based on the experimental result representing the size- and morphology-dependent lateral focusing of biological species and applied spherical model particles, the equivalent diameter of the target cells could be estimated, which can help us to predict their behavior in a hydrodynamically driven microfluidic system. The relative lateral (fluorescent and scattered light) intensity scans in Figure 6 represent the relative particle distributions at the outlet curve of the H25_W_cr_50 channel structure for *E. coli* (Figure 6A), RBC (Figure 6B), yeast (Figure 6C), and HeLa (Figure 6D) target cells and the applied model particles. Based on the comparison of the lateral-focusing positions or the extent of the concentrated regions of the biological targets and the rigid spherical beads, the behavior-equivalent spherical particle could be defined with the diameters of Ø 1.1 μm, Ø 6.08 μm, Ø 10.2 μm, and Ø 15.8 μm for the *E. coli*, RBC, yeast, and HeLa cells, respectively. It should be noted that RBC and yeast cells reached their equilibrium-focusing positions almost at the same lateral regime; accordingly, the yeast cells can be applied adequately in preliminary experiments to simulate the hydrodynamic behavior of these human cells in microfluidic flows [33].

## 5. Conclusions

The phenomenon of Dean-flow-affected lateral focusing was investigated in a periodically inhomogeneous curvilinear microfluidic channel with special focus on the behavior of biological target cells, since several biomedical applications are intended to position or separate these species by their size and morphology in flow.

Primarily, the channel-geometry- and flow-rate-dependent formation of Dean vortices was simulated by finite element modelling applying COMSOL Multiphysics FEM software. The flow-rate-dependent migration of the vortex centers was determined for a better comprehension of the evolution of the equilibrium-focusing states of particles in a microfluidic environment. Based on the numerical calculations for flow velocity distributions, the size dependent particle movements were also estimated to predict their lateral-focusing positions at the outlet of the microfluidic system using the particle-tracing unit of the COMSOL program.

To validate our theoretical results, the tracing behavior of real biological cells and polystyrene beads with different sizes was studied in variously parameterized microfluidic systems. A comprehensive map of equilibrium lateral-focusing positions were determined for rigid polystyrene beads having a diameter in the range of Ø 0.5–16.5 μm, and for representative biological cells such as *E. coli* bacteria, yeast, red blood cells, and HeLa cancer cells. The recorded trajectories of the analysed beads and cells are compared in the movie titled A_Banyai_LatFocus_Experimental.mp4 in the Appendix A. Compared to rigid spherical beads, these cells can adopt different forms of movement in flow due to their various degrees of flexibility, multi-dimensional sizes, and shapes, which can modify their lateral-focusing position. This study suggested that rigid particles can serve as an adequate model for individual biological species and an equivalent diameter can be defined for a prediction of the lateral-focusing positions of the analyzed cells in the specific microfluidic structure. Figure 7 represents the evolution of focusing positions in the investigated microfluidic channel for different spherical-bead diameters and demonstrates the behaviors of the applied target cells.

To validate the reliability of the FEM simulation, a probable focusing range in the cross-section of the channel was defined and compared to the measured focusing positions in the case of bead diameters of 1.1 µm, 6.8 µm, and 10.2 µm, respectively. The modelled probable focusing ranges are also indicated in Figure 7. The central positions of the modelled particle distribution were in accordance with the experimental case. The modelled spread of the particles—variation in the numerical simulation—was higher due to the limited number (1000) of the test particles.

In the literature review, various powerful microfluidic-cell-separation solutions [4,18,29] were presented. Their channel geometry and cross-sectional parameters mainly determine the possible cell sizes and morphologies to be separated and focused. Accordingly, the feasibility of a developed separation method depends on the compatibility of the applied biological sample and the designed microfluidic system. The crucial aspects of their applicability are the prevention of cell deterioration caused by evolving shear forces at higher flows and the inhomogeneity of the cell population affecting the success of its lateral focusing. In the case of larger HeLa cells, a much larger size distribution and shape variability were observed, so pre-filtration or an additional separation method could be necessary. The aim of the present work was to highlight the importance of the target morphology regarding the efficiency of the separation.

The comprehension of the positioning, separation, or filtration of cells in microfluidic systems was the motivation of the domestic project “Rapid urine bacteria analyzer—RUBA”, which aimed to develop a point-of-care (PoC) device to detect and analyze pathogenic bacteria or blood cells in urine. Due to the low concentration of bacteria in a sample, the efficient concentration of the target cells is crucial. In urine, squamous cells, fungi, sperm, and blood can be found in various sizes among the shaped elements, with particles within the 1–60 µm size range. Accordingly, various filtration and separation methods were tested and investigated, such as weir-type cross-flow filtration, deterministic lateral displacement (DLD), and inertial lateral focusing in the presented asymmetric curved serpentine structure. The interesting behavior of the different cells (*E. coli*, RBC, and spherical cells such as yeasts and HeLa) in a microfluidic channel was tested to understand the effects of size and shape. The proper combination of the individual structural microfluidic elements requires serious consideration, although this could provide a suitable solution for fine-tuning the filtration process. Our results could be utilized in the development of such point-of-care devices as that in the focus of 77 Elektronika Ltd. and, moreover, in promising future applications such as the separation and capturing of circulating tumor cells (CTCs).

## Figures and Tables

**Figure 1 sensors-23-00800-f001:**
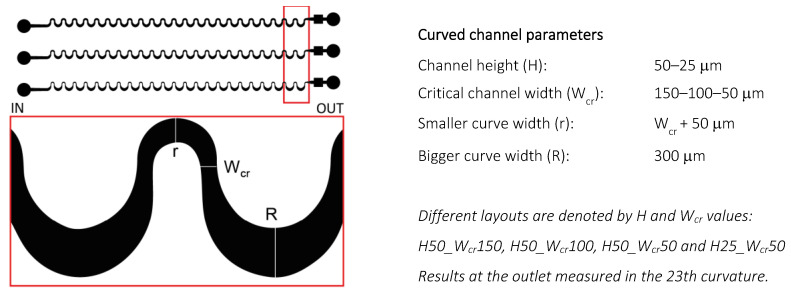
The schematic design of the lithographic mask representing the obtained geometrical parameters of the periodic sequence of the microchannels. The red box indicates the magnified area of the mask layout.

**Figure 2 sensors-23-00800-f002:**
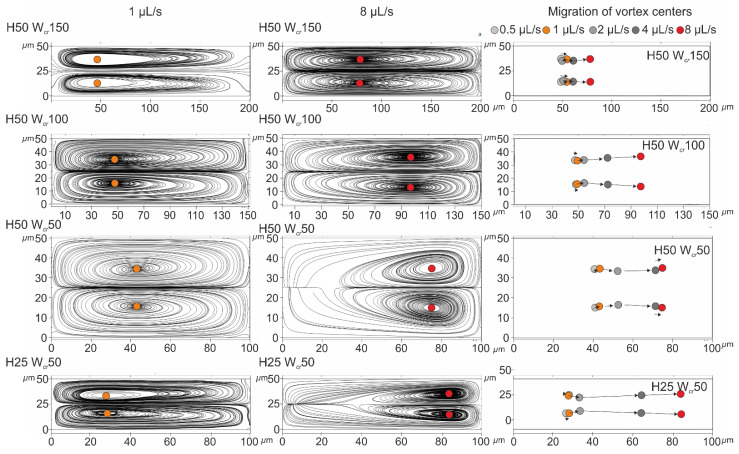
The lateral movement of the vortex’s center was followed in case of different channel parameters (H: 50 and 25 µm, W_cr_: 150, 100 and 50 µm) at 0.5, 1, 2, 4 and 8 µL/s flow rates. Velocity-field-based streamlines were visualized in the y-z plane of the channel cross-section at a smaller curvature (W_cr_ + 50 µm) at 1 and 8 µL/s flow rates.

**Figure 3 sensors-23-00800-f003:**
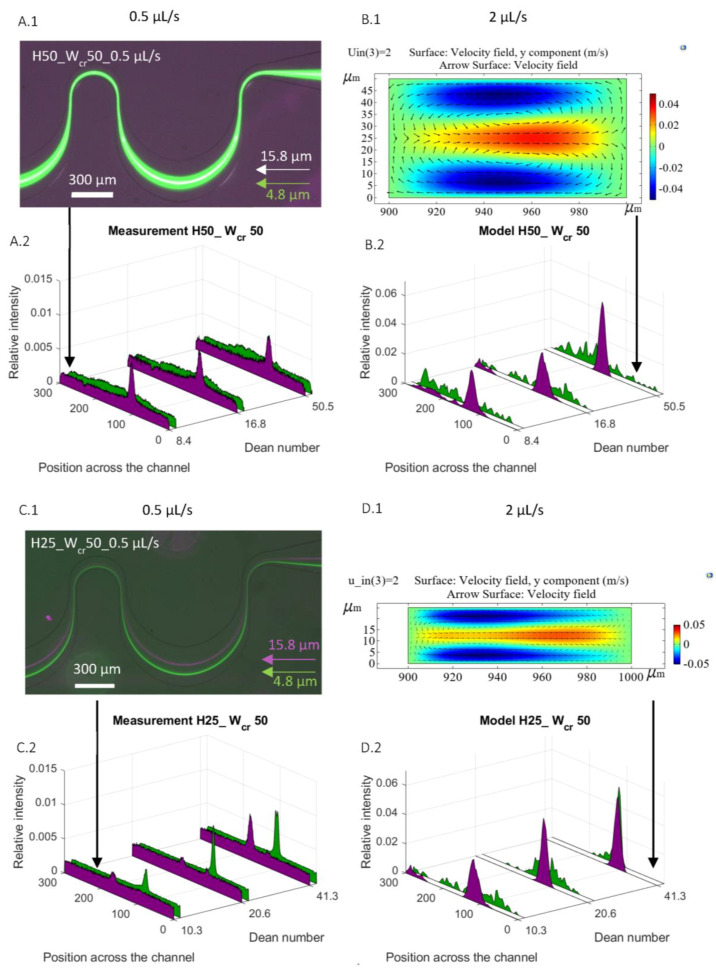
Comparison of the lateral focusing of rigid polystyrene beads (Ø 4.8 µm and Ø 15.8 µm) in periodic curved channels with the same critical width and different channel heights (W_cr_ = 50 μm; H = 50 and 25 μm) at flow rates of 0.5, 1, 2 μL/s. Experimental results (**A.1**,**A.2**,**C.1**,**C.2**) are visualized by the fluorescence-intensity-based lateral distribution of the beads detected in the bigger curve (in 300 μm cross-sectional width). (**A.1**) shows the lateral focusing position of the selected beads in channel H50_W_cr_50; (**C.1**) in channel H25_W_cr_50, both at 0.5 µL/s flow rate; and the experimental results are summarized respectively in the same channel represented on a graph at 0.5, 1, 2 µL/s (**A.2**,**C.2**). Numerical results (**B.1**,**B.2**,**D.1**,**D.2**) are demonstrated by the velocity field distribution in the smaller curve (in W_cr_ + 50 μm cross-sectional width): (**B.1**) in channel H50_W_cr_50, (**D.1**) in channel H25_W_cr_50, both at 2 μL/s flow rate; and respectively in the same channel the lateral distribution of the beads detected in the bigger curve (in 300 μm cross-sectional width) at 0.5, 1, 2 µL/s (**B.2**,**D.2**).

**Figure 4 sensors-23-00800-f004:**
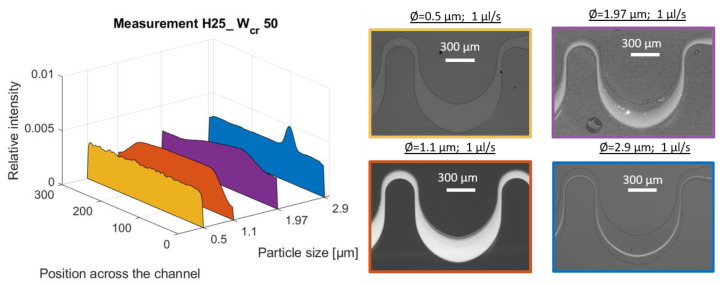
The size limit of lateral focusing in channel H25_W_cr_50 at 1µL/s. The applied bead sizes are Ø = 0.5, 1.1, 1.97 and 2.9 µm.

**Figure 5 sensors-23-00800-f005:**
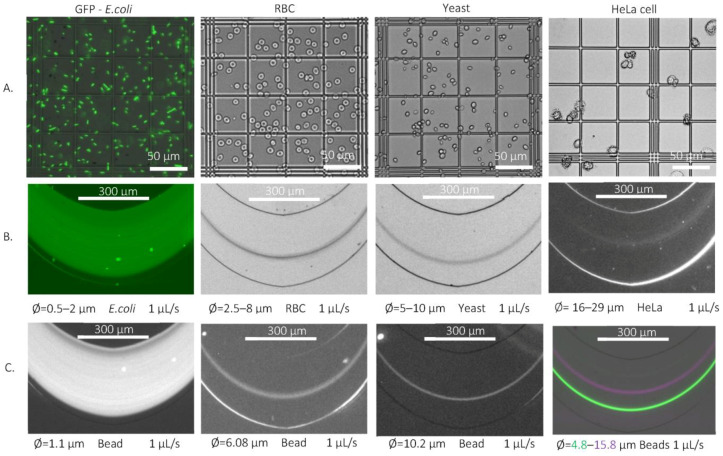
Lateral focusing of biological cells and their rigid model beads in channel H25_W_cr_50 at 1 µL/s flow rate. The size and morphology of the applied biological sample were studied in a Bürker chamber (**A**), and their lateral positions were captured by fluorescent or dark-field microscopy at the end of the channel in the big curvature (**B**). The equivalent diameter of the studied model beads was determined by the maximal overlapping of the lateral-focusing positions compared to the given cell. (**C**) is the polystyrene bead profiles that best describes the lateral focusing position and profile of the given cell sizes.

**Figure 6 sensors-23-00800-f006:**
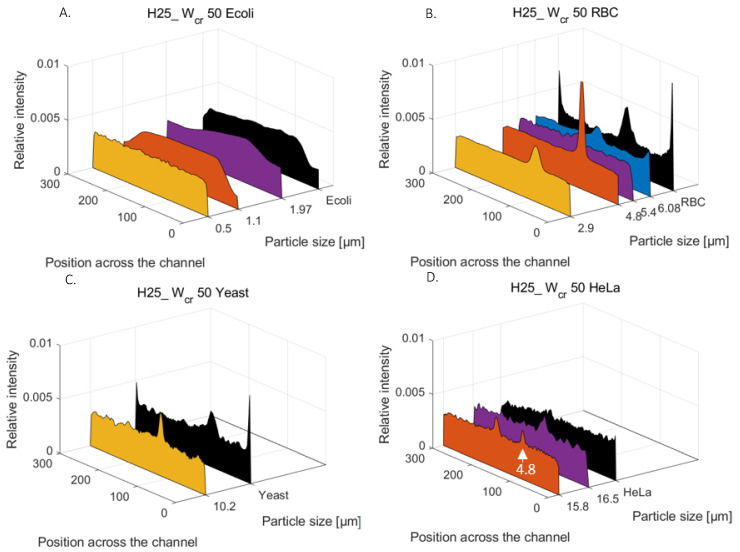
Experimentally determined lateral distributions of the studied biological cells and their model beads developed in the periodically inhomogeneous curvilinear-focusing microfluidic channel in H25_W_cr_50 at 1 µL/s flow rate: (**A**) for *E. coli* with polystyrene beads of Ø 0.5, 1.1 and 1.97 µm; (**B**) for Red blood cells (RBC) with beads of Ø 2.9, 4.8, 5.4 and 6.08 µm; (**C**) Saccharomyces cerevisiae (Yeast) with bead of Ø 10.2 µm; (**D**) HeLa cells with polystyrene beads of Ø 4.8, 15.8 and 16.5 µm.

**Figure 7 sensors-23-00800-f007:**
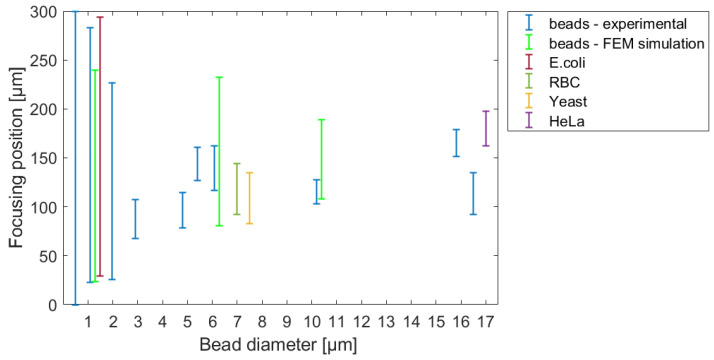
The focusing positions and concentration ranges of rigid beads and target cells in the applied lateral-focusing microfluidic system (projected to a demonstrative 300 μm wide channel cross-section). The modelled probable focusing ranges are also indicated in the case of bead diameters of 1.1 µm, 6.8 µm, and 10.2 µm, respectively.

**Table 1 sensors-23-00800-t001:** Mesh-dependent convergency study. Results converged after Mesh #5; mesh #7 was selected to achieve the optimal runtime and accuracy.

Mesh	Mesh #1	Mesh #2	Mesh #3	Mesh #4	Mesh #5	Mesh #6	Mesh #7
Avg. vorticity on the probe surface (1/s)	9532.1	9425.3	9394.8	9353.7	9366.7	9365.7	9365.3
No. of mesh elements	35,078	63,664	159,509	309,853	848,159	2,468,031	2,523,010

**Table 2 sensors-23-00800-t002:** The studied biological cells and the compared model bead sizes for lateral focusing in channel H25_W_cr_50.

			Size		
	Used Cell	Morphology	Height	Width	Size of the Model Beads (Ø)
Bacteria [34]	*E. coli*	stick	0.5 µm	2 µm	0.5, 1.1, 1.97 µm
Hematopoietic [35,36]	Red blood cell	discoid biconcave	2.5 µm	6–8 µm	2.9, 4.8, 5.4, 6.08 µm
Yeast [37]	Saccharomyces cerevisiae	round or oval	5–10 µm		5.4, 6.08, 10.2 µm
Cancer cell line [38]	HeLa cell	diverse, inhomogeneous, and spherical in suspension	16–29 µm		15.8, 16.5 µm

**Table 3 sensors-23-00800-t003:** The *a/D_H_* value calculated from the particle and channel size ratio characterized the criteria of size-dependent lateral focusing in the case different biological targets and model beads. The experimental focusing position (or concentration range) is also summarized. Gray background marks the cases where lateral focusing nor theoretically nether experimentally can occur in the microfluidic channel.

H25_Wcr50 (1 µL/s)	Particle Diameter (a) (µm)	a/D_H_	Fluorescent Band Position (0–300 µm)	Intensity Peak at (µm)
Polysterene beads	0.5	0.015	[0–300]	-
	1.1	0.033	[22.6–283]	-
	1.97	0.059	[25.5–226.4]	-
	2.9	0.087	[67.9–107.5]	90
	4.8	0.144	[78.6–114.3]	98.2
	5.4	0.162	[126.8–160.8]	138
	6.08	0.182	[117–162]	135
	10.2	0.306	[103.03–127.3]	117
	15.8	0.474	[151.5–178.6]	164.2
	16.5	0.495	[91.8–134.7]	114
*E. coli*	0.5	0.015	[28.37–294.3]	-
	2.0	0.060		
RBC	2.5	0.075	[91.8–143.9]	126
	8.0	0.240		
Yeast	5.0	0.150	[82.7–134.7]	105
	10.0	0.300		
HELA	16.0	0.480	[162.4–197.9]	174
	29.0	0.870		

## Data Availability

Not applicable.

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
