# Peer review of "Dean-Flow Affected Lateral Focusing and Separation of Particles and Cells in Periodically Inhomogeneous Microfluidic Channels"

_sensors, 2023, doi:10.3390/s23020800_

Round 1
Reviewer 1 Report
The manuscript “Dean-Flow Affected Lateral Focusing and Separation of Particles and Cells in Periodically Inhomogeneous Microfluidic Channels” presents a comprehensive study of particle focusing in asymmetric serpentine structures with a focus on size and form of microparticles and cells. As a result of this study, the authors propose equivalent bead sizes to describe morphology effects on the lateral focusing position. The manuscript has a comprehensible structure and descriptions of the applied methods. The authors provided a thorough theoretical and literature background and discussion. They applied finite element as well as microscopy-based imaging to strengthen their study.
Overall, this is a nice paper and in general I recommend the publication of the manuscript. However, I wonder if "Sensors" is the appropriate journal, as this study does not deal with sensor development / sensing technologies and therefore, in my opinion, does not fit into the scope of the journal.
The reviewer has further general comments to the author:
- Can cell suspensions also be separated from each other with the device? Since there is great interest in separating complex cell samples, another experiment in which the different cells (E.coli, RBC, Yeast, HeLa) in a sample are mixed and separated from each other should be shown.
- Table 1: Please add references for cell sizes.
- Table 2 seems to be screenshot of a figure and appears quite blurry.
- The authors should also refer to one of the most recent papers on CHO cell separation using a 3D-printed spiral separator by Enders et al. in their manuscript, as this work is highly relevant to the topic. "3D Printed Microfluidic Spiral Separation Device for Continuous, Pulsation-Free and Controllable CHO Cell Retention" https://doi.org/10.3390/mi12091060
- In general, there are spelling mistakes such as “dropplets” or missing spaces etc. Please check and revise carefully.
- I would greatly recommend the numbering of the equations.
Author Response
Answer to REVIEWER1
regarding A. Bányai et al. Dean-Flow Affected Lateral Focusing and Separation of Particles and Cells in Periodically Inhomogeneous Microfluidic Channels
We would like to thank the reviewer his advanced comments. Our motivation to publish our results in the Sensors Journal is twofold. Primarily we believe that our research could support the development of cell analytical devices dedicated to detect and analyse pathogen bacteria or blood cells – we work together with 77 Elektronika Ltd. on these applications. Moreover our previous publication in the topic ( https://doi.org/10.3390/s22093474 ) published in the Special Issue "MEMS Devices for Biomedical Applications" of MDPI Sensors. This paper will summarise our novel results on the topic focusing on the lateral focusing of real cells compared to particles having different sizes. We would be appreciated in case work could appear continuously in the same Journal and would be helpful to better understanding the phenomenon and applicability of inertial migration.
Our responses to the comments:
I.1. Can cell suspensions also be separated from each other with the device? Since there is great interest in separating complex cell samples, another experiment in which the different cells (E.coli, RBC, Yeast, HeLa) in a sample are mixed and separated from each other should be shown.
We proved that size dependent separation of rigid particles and cells from each other is possible. As a model, fluorescent polystyrene beads were used to clearly demonstrate the focused states of a given cell size within the microfluidic channel, and how the morphology of the various cells affects this focusing position. In our previous work, we were able to separate beads with a diameter of 4.8 and 15.8 µm from the same suspension, there the different fluorescent labelling allowed us to clearly identify the particle size from a suspension ( https://doi.org/10.3390/s22093474 ). The separation of these bead sizes is also shown in Figure 3 considering the geometry of the microfluidic channel. However, the visualisation the separation of unlabelled real cells is a challenge considering the limited applicability of darkfield microscopy and its need for highly sensitive and high speed imaging of biological sample.
I.2. Table 1: Please add references for cell sizes.
The sources of the cell sizes were indicated in the literature review for the given cell, but of course this was also added in the table. Designation of references for cell size: RBC [33, 34]; E.coli [32]; Yeast cell [35]; HeLa cell [36] (Note, this table is Table 2 after revision.)
I.3. Table 2 seems to be screenshot of a figure and appears quite blurry.
Table 2 was changed in the final document.
I.4. The authors should also refer to one of the most recent papers on CHO cell separation using a 3D-printed spiral separator by Enders et al. in their manuscript, as this work is highly relevant to the topic. "3D Printed Microfluidic Spiral Separation Device for Continuous, Pulsation-Free and Controllable CHO Cell Retention" https://doi.org/10.3390/mi12091060.
Thank you for your comment, we have processed this relevant and useful article, and we marked the article in our references also.
I.5. In general, there are spelling mistakes such as “dropplets” or missing spaces etc. Please check and revise carefully.
Thank You for the notice, we also checked the manuscript more thoroughly and made it reviewed by English expert to bring it to a more acceptable level.
I.6. I would greatly recommend the numbering of the equations.
This was a shortcoming of our final processing, the numbering of the equations were indicated.
Thank you for your comments and constructive criticism!
Yours Sincerely,
the authors of this article

Reviewer 2 Report
This research paper describes
The purpose of the recent work is to give a better explanation, how Dean vortices affect lateral focusing, and to understand how cell morphology can alter focusing position compared to spherical particles. Position and extent of the focused region were investigated using polystyrene fluorescent beads with different bead diameters (Ø = 0.5 - 1.1 - 1.97 - 2.9 - 4.8 - 5.4 - 6.08 - 10.2 - 15.8 - 16.5 µm) at different flow rates (0.5 – 1 - 2 µL/s). Size-dependent focusing generates a precise map of the equilibrium positions of the spherical beads at the end of the analysed periodically altering channels, which gave a good benchmark for focusing multi-dimensional particles and cells. Biological samples used for experiments were rod-shaped E.coli, donut-shaped Red Blood Cells (RBC), round or ovoid-shaped yeast, Saccharomyces cerevisiae and soft-irregular shaped HeLa cancer cell-line cells to understand how the shape of the cells affects the focusing position at the end of the channel.
1. Figure 2, 3 Quality needs improvement
2. Physics equations and boundary conditions need to be added
3. For validation of results simulation and experimental results need to be compared along with %age error calculations
4. For the novelty of work verification results must be compared with literature along with references
5. Mesh or Grid independence is missing and mesh statics as well for the accuracy of results
6. Fabrication of channel and experimental procedure and setup needs to be included in the paper
7. Supplementary file videos will help reader for better understanding of work
8. Application of work or future work must be added.
Author Response
Answer to REVIEWER2
regarding A. Bányai et al. Dean-Flow Affected Lateral Focusing and Separation of Particles and Cells in Periodically Inhomogeneous Microfluidic Channels
We would like to thank the reviewer for advanced comments, and constructive criticism!
Our responses to the comments:
II.1. Figure 2, 3 Quality needs improvement
Thank you for your comment, we improved the quality of the marked images and the visibility of the axis labels. Figure 3 was reorganised also.
II.1. Physics equations and boundary conditions need to be added
The boundary conditions were detailed in the 3.4. section: “At the inlet a laminar inflow boundary was applied with varying flow rates between 0.5 and 6 μL/s. Zero pressure with suppressed backflow was set as outlet boundary condition and ‘no slip’ condition was set for the channel walls.” Comprehensive description of the model calculations were upgraded and a mesh convergence study was also added to the manuscript – see Table 1.
II.3. For validation of results simulation and experimental results need to be compared along with %age error calculations.
We can hardly compare the model to the experimental results, due to in model calculations only 1000 particles were used to define the theoretical equilibrium positions of the test beads and these were validated by the real particle focusing trajectories demonstrated by the fluorescent distributions. Although we could define the probable focusing range in the cross section and this range could be compared to the measured focusing positions. Please find the modified Figure 7, where the modelled probable focusing ranges were also added in case of bead diameters of 1.1µm, 6.8µm and 10.2µm, respectively.
Figure 7. The focusing positions and concentration ranges of rigid beads and target cells in the applied lateral focusing microfluidic system (projected to a demonstrative 300 µm wide channel cross section). The modelled probable focusing ranges were also indicated in case of bead diameters of 1.1µm, 6.8µm and 10.2µm, respectively.
II.4. For the novelty of work verification results must be compared with literature along with references
Conclusion section was extended according to these requirements.
II.5. Mesh or Grid independence is missing and mesh statics as well for the accuracy of results.
The independence study of the mesh has been added to section 3.4. Comprehensive description of the model calculations were upgraded and a mesh convergence study was also added to the manuscript – see Table 1. Accuracy of the results was also referred at adding the global error estimate value of the model.
II.6. Fabrication of channel and experimental procedure and setup needs to be included in the paper.
In paragraph 3.2 the manufacturing and in paragraph 3.5 the experimental arrangement were detailed and highlighted in the resubmitted manuscript, references were also indicated.
II.7. Supplementary file videos will help reader for better understanding of work.
Video recording of the cell movement is quite challenging, due to the low fluorescent or scattered light intensities – this would need extreme sensitive imaging methods. Although a special movie was prepared about the alteration of the focusing trajectories in the last bend of the periodically curved channel system as the function of the bead and cell sizes. The lateral position of the trajectories were recorded at 1 µL/s flow rate at the end of the fluidics, in the 23rd, large curvature. The images were taken with a long exposure time in fluorescent microscopy mode in case of polystyrene beads and in dark field scattered light mode in case of non-labelled cells. The movie could be applied as supplementary.
Other animation was created from the simulations, which can represent the focusing process in a periodic microfluidic system containing 100 curves. In this case a larger channel cross-section ( H50 Wcr150 ) were applied with 3µL/s flow rate, and two different bead sizes – 4.8 µm and 15.8 µm bead diameters. The continuous movement of the 1000 bead was visualised by a Poincare map from bend to bend, as a representation the approach of their size dependent equilibrium positions in the Dean vortices developed along the entire length of the microfluidics. This animation could also be applied as supplementary.
II.8. Application of work or future work must be added.
The positioning, separation or filtration of cells in microfluidic systems were motivated by the domestic project “Development of rapid urine bacteria analyser – RUBA” aimed to detect and analyse pathogenic bacteria of blood cells in urine. Due to the low concentration of bacteria in the sample the efficient concentration of the target cells is crucial. Accordingly various filtration and separation methods were tested and investigated, as weir-type crossflow filtration, deterministic lateral displacement – DLD and inertial lateral focusing. The interesting behaviour of the different cells (E.coli, RBC, spherical cells as yeasts and HeLa) in the microfluidics were tested to understand the effects of size and shape. Our results could be utilised in the development of such Point-of-Care devices being in the focus of the 77 Elektronika Ltd. A short highlight of future applications were inserted in the Conclusions section.
Thank you for your comments and constructive criticism!
Yours Sincerely,
the authors of this article

Reviewer 3 Report
The authors of the manuscript titled "Dean-Flow Affected Lateral Focusing and Separation of Particles and Cells in Periodically Inhomogeneous Microfluidic Channels" explains how dean vortex affects literal focusing, thus further expanding its discovery in the use of understanding cell morphology alteration compared to spherical particles, which is a novel approach by its own. To understand how the shape of the cells affects the focusing position at the end of the channel, rod-shaped E. coli, donut-shaped Red Blood Cells (RBCs), round or ovoid-shaped yeast, Saccharomyces cerevisiae, and soft-irregular-shaped HeLa cancer cells-lines were used as samples for experiments by the authors.
However the novelty of the work is excellent, but minor amendments are needed before it can be accepted for publication.
1) Why the authors chose "Position and extent of the focused region were investigated using polystyrene fluorescent beads with different bead diameters (Ø = 0.5 - 1.1 - 1.97 - 2.9 - 4.8 - 5.4 - 6.08 - 10.2 - 15.8 - 16.5 μm) at different flow rates (0.5 – 1 - 2 μL/s)." as experimental conditions?
2) Chapter 2: Theoretical Background the mathematical equations do not have any reference. That's quite odd. The authors need to add proper referencing to clarify the theory. For example, Yin et al.l used "Wave-shaped microfluidic chip assisted point-of-care testing for accurate and rapid diagnosis of infection,s" a similar theoretical approach to establish the theory that can be seen as an example. The groups have been pioneering dean vortex and particle analysis for many years and can be taken to embellish the approach involving dean vortex.
3) From the microfluidic aspect, why the authors used this specific model? Is it based on the theoretical relationship they established before from chapter 2? Which lacks proper reference.
4) Based on the numerically calculated flow velocity distributions, the size-dependent particle movements were also estimated to predict their lateral focusing positions at the outlet of the microfluidic system using the Particle Tracing unit of the Comsol code.
Here is what the authors meant by "code."
5) A thorough grammar check is needed before the next submission.
6) Easier words can be used rather than specific words in general.
7) The whole manuscript is suggested to use of a proper manner of the usage of passive and active voice. The clustered use of passive and active voice mixers confuses the readers about the author's intention.
Author Response
Answer to REVIEWER3
regarding A. Bányai et al. Dean-Flow Affected Lateral Focusing and Separation of Particles and Cells in Periodically Inhomogeneous Microfluidic Channels
We would like to thank the reviewer his advanced comments, and constructive criticism!
Our responses to the comments:
III.1. Why the authors chose "Position and extent of the focused region were investigated using polystyrene fluorescent beads with different bead diameters (Ø = 0.5 - 1.1 - 1.97 - 2.9 - 4.8 - 5.4 - 6.08 - 10.2 - 15.8 - 16.5 μm) at different flow rates (0.5 – 1 - 2 μL/s)." as experimental conditions?
In our previous work ( https://doi.org/10.3390/s22093474 ), we carried out focusing experiments with Ø = 4.8 and 15.8 µm beads with intensive channel parameterization and definition of the adequate channel cross-section in which both bead sizes can be focused laterally in a suspension. We further tested the proven channel H25_Wcr50 to explore, what is the smallest bead size besides we can still achieve lateral focusing. The motivation was to define a reliable geometry and flow parameters for focusing real cells. Due to the targeted cell sizes are between 0.5µm (E.coli) and 10 – 20µm (blood cells or tumor cells - CTCs) we decided to test polystyrene beads as equivalent models and to choose bead sizes that covers this range. The flow rates were defined to ensure adequate flow conditions for inertial focusing – without deteriorating the cell membranes caused by the share forces emerging in case of high flow velocities.
III.2.Chapter 2: Theoretical Background the mathematical equations do not have any reference. That's quite odd. The authors need to add proper referencing to clarify the theory. For example, Yin et al.l used "Wave-shaped microfluidic chip assisted point-of-care testing for accurate and rapid diagnosis of infections" a similar theoretical approach to establish the theory that can be seen as an example. The groups have been pioneering dean vortex and particle analysis for many years and can be taken to embellish the approach involving dean vortex. ”
Thank you for your recommendation. Previously we tried to mark the most important references at the beginning of the chapter where we mentioned the principal equations, although in the revised version we clarified the theoretical background more precisely. According to the request, we inserted another references that summarizes the most important equations, which is relevant to our topic [1]:
[1] ‘Micromixers - 2nd Edition’. https://www.elsevier.com/books/micromixers/nguyen/978-1-4377-3520-8 (accessed Dec. 18, 2022).
III.3. From the microfluidic aspect, why the authors used this specific model? Is it based on the theoretical relationship they established before from chapter 2? Which lacks proper reference.
We use Comsol because while other flow simulation softwares use a finite volume method for CFD modelling, Comsol's solver uses the finite element method and this helps to integrate different physical models (e.g. diffusion, thermal propagation, magnetic field). Furthermore, the program can be perfectly parameterized and the execution can be automated by the help of Matlab code.
III.4. Based on the numerically calculated flow velocity distributions, the size-dependent particle movements were also estimated to predict their lateral focusing positions at the outlet of the microfluidic system using the Particle Tracing unit of the Comsol code. Here is what the authors meant by "code."
Thank you for your comment, probably the correct word is “program” instead of “code”.
III.5.- III.6.- III.7. A thorough grammar check is needed before the next submission.”- “Easier words can be used rather than specific words in general.” - “The whole manuscript is suggested to use of a proper manner of the usage of passive and active voice. The clustered use of passive and active voice mixers confuses the readers about the author's intention.”
We received similar comments from the other reviewers. The English spelling and wording truly leave something to be desired. We checked again the manuscript more thoroughly to bring it to a more acceptable level before resubmission.
Thank you for your comments and constructive criticism!
Yours Sincerely,
the authors of this article

Round 2
Reviewer 2 Report
All the comments are addressed properly, and I recommend the paper for possible
publication in this journal after the following minor changes.
In mesh independence/ conversion paragraph the following reference need to be added.
Numerical analysis of non-aligned inputs M-type micromixers with different shaped obstacles for biomedical applications
Author Response
Answer to REVIEWER2
regarding A. Bányai et al. Dean-Flow Affected Lateral Focusing and Separation of Particles and Cells in Periodically Inhomogeneous Microfluidic Channels
We would like to thank the reviewer for further comment. In accordance with the request, we replaced the missing reference, which can be found under number [31].
Thank you for your comments and constructive criticism!
Yours Sincerely,
the authors of this article